# SUCNR1 Mediates the Priming Step of the Inflammasome in Intestinal Epithelial Cells: Relevance in Ulcerative Colitis

**DOI:** 10.3390/biomedicines10030532

**Published:** 2022-02-24

**Authors:** Cristina Bauset, Lluis Lis-Lopez, Sandra Coll, Laura Gisbert-Ferrándiz, Dulce C. Macias-Ceja, Marta Seco-Cervera, Francisco Navarro, Juan V. Esplugues, Sara Calatayud, Dolores Ortiz-Masia, Maria D. Barrachina, Jesús Cosín-Roger

**Affiliations:** 1Departamento de Farmacología, Facultad de Medicina, Universidad de Valencia, 46010 Valencia, Spain; cristina.bauset@uv.es (C.B.); lluislis@alumni.uv.es (L.L.-L.); sandra.coll@uv.es (S.C.); laura.gisbert@uv.es (L.G.-F.); macias.dcc@gmail.com (D.C.M.-C.); juan.v.esplugues@uv.es (J.V.E.); sara.calatayud@uv.es (S.C.); dolores.barrachina@uv.es (M.D.B.); 2Hospital Dr. Peset, FISABIO, 46017 Valencia, Spain; marta.seco@uv.es; 3Hospital de Manises, 46940 Valencia, Spain; fran.navarro.vicente@gmail.com; 4CIBERehd, 28029 Madrid, Spain; 5Departamento de Medicina, Facultad de Medicina, Universidad de Valencia, 46010 Valencia, Spain

**Keywords:** Ulcerative Colitis, SUCNR1, epithelial cells

## Abstract

Intestinal epithelial cells (IECs) constitute a defensive physical barrier in mucosal tissues and their disruption is involved in the etiopathogenesis of several inflammatory pathologies, such as Ulcerative Colitis (UC). Recently, the succinate receptor SUCNR1 was associated with the activation of inflammatory pathways in several cell types, but little is known about its role in IECs. We aimed to analyze the role of SUCNR1 in the inflammasome priming and its relevance in UC. Inflammatory and inflammasome markers and SUCNR1 were analyzed in HT29 cells treated with succinate and/or an inflammatory cocktail and transfected with SUCNR1 siRNA in a murine DSS model, and in intestinal resections from 15 UC and non-IBD patients. Results showed that this receptor mediated the inflammasome, priming both in vitro in HT29 cells and in vivo in a murine chronic DSS-colitis model. Moreover, SUNCR1 was also found to be involved in the activation of the inflammatory pathways NFкB and ERK pathways, even in basal conditions, since the transient knock-down of this receptor significantly reduced the constitutive levels of pERK-1/2 and pNFкB and impaired LPS-induced inflammation. Finally, UC patients showed a significant increase in the expression of SUCNR1 and several inflammasome components which correlated positively and significantly. Therefore, our results demonstrated a role for SUCNR1 in basal and stimulated inflammatory pathways in intestinal epithelial cells and suggested a pivotal role for this receptor in inflammasome activation in UC.

## 1. Introduction

Intestinal epithelial cells (IECs) are pivotal cells in the maintenance of mucosal integrity. They do so by forming a physical barrier, acting as sensors for damage- or pathogen-associated molecular patterns and regulating immune cells [1]. The activation of the pattern recognition receptors in these cells stimulates several intracellular signaling pathways, such as NFкB and MAPKs pathways, which, in turn, induce the release of chemokines and interleukins [2]. In the gastrointestinal tract, the disturbance of the epithelial barrier allows for interaction between microbiota, dietary antigens and other compounds with the immune system, which have been involved in the etiopathogenesis of Inflammatory Bowel Disease (IBD) [3]. Ulcerative Colitis (UC) is a subtype of IBD characterized by a diffuse, continuous and chronic inflammation limited to the mucosa and submucosa layers of the colon. Given the increasing incidence and prevalence in both developed and developing countries, UC has become a public health challenge worldwide [4] whose etiology is widely unknown [5,6]. Among all the molecular pathways involved in the activation of inflammation described so far, the inflammasome complex has recently come to be considered a critical regulator of intestinal homeostasis [7,8].

Inflammasomes constitute a group of cytosolic protein complexes whose main function is to recognize exogenous microbes and danger-associated endogenous components. The activation of these complexes triggers the proteolytic cleavage of pro-Caspase-1 and the production of IL-1β and IL-18, which in turn, starts the inflammatory and immune responses [9,10]. Although several types of inflammasomes have been described so far, including NLRP1, NLRP2, NLRP3, NLRC4 and AIM2, the most characterized and studied has been NLRP3, due to its clinical importance in a wide range of pathologies, such as atherosclerosis, diabetes or IBD [11]. In order to induce a complete activation of the inflammasome, a mechanism of two consecutive steps occurs. During the first step, called priming, there is an induction of the transcriptional expression of the central components of the inflammasome such as NLRP3, ASC, IL-1β and IL18. Next, in the second step, called activation or triggering, the complete activity of the NLRP3 inflammasome is promoted and, subsequently, the activation of the pro-Caspase-1 and the release of IL-1β occur [12]. Of interest, a key role of NLRP3 inflammasome has been described in several chronic inflammatory pathologies, including UC. In fact, the importance of NLRP3 in UC patients has been demonstrated by the presence of some SNPs in NLRP3 gene, such as rs10754558 and rs10925019, which are associated with a higher susceptibility of this pathology [13]. However, the specific role of NLRP3 in UC is still unclear, since controversial results have been reported in murine models of colitis. Some studies in NLRP3 knock-out mice showed protection in TNBS, DSS and oxazolone models of colitis [14,15,16], whereas others have described a detrimental or worse colitis in the same knock-out mice [17,18,19]. These discrepancies have been explained by two theories: the differences in the intestinal microbiota of these mice [20] or the cell-specific role of NLRP3 inflammasomes. In fact, while, in epithelial cells, it should help in the maintenance of homeostasis, its activation in immune cells might have a harmful effect [21]. Therefore, further studies are required in order to better elucidate the specific role of NLRP3 in the pathogenesis of UC and identify the pathways that prime NLRP3 inflammasomes. 

In recent years, succinate, an intermediate metabolite of the tricarboxylic acid (TCA) cycle, acting through the G-protein-coupled receptor SUCNR1, has been involved in several inflammatory and metabolic pathologies [22,23,24,25,26]. This receptor, initially named GPR91, is expressed in several cell types, such as hepatic stellate cells, cardiomyocytes, platelets, dendritic cells, macrophages, epithelial cells and fibroblasts, and is found in several tissues, including liver, heart, kidney, retina, immune system and the gastrointestinal tract [27]. The binding of succinate to SUCNR1 triggers occurs as follows: the displacement of the GDP by GTP, the release of the α subunit and the βγ dimer and the induction of different downstream signaling pathways. The activation of one signaling pathway via both α and/or βγ subunits seems to be cell and tissue-specific; in cardiomyocytes, the α subunit increases the cAMP levels and activates the PKA, whereas in adipose tissue, it reduces the cAMP levels. Regarding the βγ dimer, it activates the PI3K-Akt-Src in platelets, while it phosphorylates ERK-1/2 in dendritic cells [28]. The specific role of this receptor in immune cells has been extensively analyzed and both proinflammatory and anti-inflammatory effects have been reported [29]. On the one hand, SUCNR1 amplifies the release of IL-1β from macrophages [30], enhances the chemotaxis in white adipose tissue of monocyte-derived macrophages [31] and can also decrease the expression of IL10, TLR4 and TLR5 and increase the expression of TNF-α in peripheral blood mononuclear cells [32]. On the other hand, SUCNR1 promotes an anti-inflammatory effect through PKA–CREB–KLF4 pathways in adipose tissue macrophages [33]. Nevertheless, no information is available about the role of SUCNR1 in epithelial cells. 

In the present study, we aimed to analyze the role of SUCNR1 in the priming step of the inflammasome activation in intestinal epithelial cells, and its relevance in colitis. Results demonstrated, for the first time, that SUCNR1 mediated NLRP3 priming in both intestinal epithelial cells and a murine model of DSS-colitis, and that this receptor correlated with inflammasome markers in human UC.

## 2. Materials and Methods

### 2.1. Patients

Intestinal resections from UC patients with severe refractory disease state who underwent surgery were obtained. In the case of non-IBD patients, nondamaged mucosa of colonic resections from patients with colorectal cancer were used as controls. The information on all the patients analyzed in this study is summarized in Table 1. The study was approved by the Institutional Review Board of the Hospital of Manises (Valencia, Spain). Written informed consent was obtained from all participating patients.

### 2.2. Mice

To perform in vivo experiments, wild type (WT) C57Bl/6 and SUCNR1^−/−^ mice (9–12 weeks old, 20–25 g weight, kindly provided by Dr. Kenneth McCreath and Dr. Ana Cervera) bred into a C57Bl/6 background [34] were used. In all cases, mice were co-housed to reduce possible differences in microbiota and maintained under a defined pathogen-free environment. The institutional animal care and use committees of University of Valencia approved all protocols. All experiments were performed in compliance with the European Animal Research Law (European Communities Council Directives 2010/63/EU, 90/219/EEC, Regulation (EC) No. 1946/2003), and Generalitat Valenciana (Artículo 31, Real Decreto 53/2013).

### 2.3. Introduction of Experimental DSS Colitis in Mice

Male 6–8-week-old C57Bl/6, and SUCNR1^−/−^ mice received vehicle or Dextran Sulfate Sodium (DSS, 40 kDa, Sigma-Aldrich, St. Louis, MO, USA) via their drinking water solution with 4 cycles of increasing percentages of DSS (1%, 1%, 1.5% and 1.5%) over 7 days, intercalated by 10 days with water. Hence, there were a total of four groups: WT vh, WT DSS, SUCNR1^−/−^ vh and SUCNR1^−/−^ DSS (*n* = 10 mice per group). Body weight and clinical signs of disease were recorded from day 1. At the end of the last cycle, on day 60, mice were properly handled and euthanized, and colon tissue samples were collected for further analysis.

### 2.4. Cell Culture

HT29 cells (American Type Culture Collection, Manassas, VA, USA) were used to perform in vitro experiments. McCoy’s Medium Modified (Sigma-Aldrich, Madrid, Spain), supplemented with 100 U/mL penicillin, 100 µg/mL streptomycin, 2 mM L-glutamine and 10% inactivated FBS, was used to culture these cells. Depending on the experiment, HT29 cells were treated for 24 h with LPS 0.1 µg/mL, succinate 1 mM, the MEK inhibitor U0126 10 µM or an inflammatory cocktail containing TNF-α 25 ng/mL, IFN-γ 20 ng/mL, LPS 1 µg/mL.

### 2.5. Small Interfering (siRNA) Transfection

HT29 cells were transfected using a control siRNA (siCtrl) and a specific SUCNR1 siRNA (Invitrogen Life Technologies, Barcelona, Spain) at a concentration of 20 pmol. Lipofectamine-2000 (Invitrogen Life Technologies, Barcelona, Spain) was also used following the manufacturer’s instructions. Gene and protein expression of SUCNR1 was analyzed by qPCR and Western Blot in order to determine the efficiency of the transfection.

### 2.6. Protein Extraction and Western Blot Analysis

Protein was isolated from HT29 cells, colons of mice and human intestinal resections, as previously described [35]. Western Blot was performed to analyze protein expression. SDS-PAGE gels were used and equal amounts of protein were loaded. Then, proteins were transferred to nitrocellulose membranes, which were further incubated with specific primary antibodies (detailed in Table 2) as well as the secondary antibodies peroxidase-conjugated anti-rabbit IgG (Thermo Scientific, Waltham, MA, USA, 1:5000) or anti-mouse IgG (Invitrogen, Waltham, MA, USA, 1:2000). Protein bands were detected with Immobilon^®^ Forte Western HRP Substrate (Millipore, Burlington, MA, USA) or Immobilon^®^ Crescendo Western HRP Substrate in AMERSHAM ImageQuant 800 (GE lifescience, Cornellà de Llobregat, Spain). To normalize protein bands, Glyceraldehyde 3-phosphate dehydrogenase (GAPDH) was used as housekeeping and Multi Gauge V3.0 software (Fujifilm Life Sciences, Cambridge, MA, USA) was used to quantify the densitometry of the bands.

### 2.7. RNA Isolation and Real-Time Quantitative PCR (RT-qPCR)

Total RNA from cells and murine and human tissues was isolated using direct-zol RNA MiniPrep Plus R2072 from ZymoResearch according to the manufacturer’s instructions. Mice tissue and intestinal resections were homogenated with TRI Reagent^®^ (ZymoResearch, Irvine, CA, USA), using the GentleMACS Dissociator (Milteny Biotech, Gladbach, Germany) as previously described [36]. cDNA was obtained from previously isolated RNA by reverse transcription PCRusing the the PrimeScript RT reagent Kit (Takara Bitechnology, Dalian, China). Gene expression was analyzed by real-time Quantitative PCR using SYBR^®^ Ex Taq (Takara Bio Inc., Saint-Germain-en-Laye, France) in LightCycler thermocycler (Roche Diagnostics, Mannheim, Germany). Specific oligonucleotides detailed in Table 3 and Table 4 were designed to perform the analysis. The relative gene expression, as fold increase, was expressed as follows: change in expression (fold) = 2 − Δ(ΔCT) where ΔCT = CT (target) − CT (housekeeping) and Δ(ΔCT) = ΔCT (treated) − ΔCT (control), where β-actin was the housekeeping gene used. 

### 2.8. Succinate Quantification

HT29 cells supernatant was used to quantify the succinate levels using the Succinate Assay Kit (Abcam, Cambridge, UK) according to the manufacturer’s instructions. Briefly, supernatants were filtered with 10 kDa spin columns (ab93349, Abcam). The Reaction Mix from the kit was used to incubate samples in 96-well plates at 37 °C during 30 min. Finally, the absorbance at 450 nm was measured with the microplate reader SpectraMax Plus 384 (Molecular Devices, San Jose, CA, USA) and the succinate concentration was calculated using the standard curve.

### 2.9. IL-1β ELISA

Secreted protein levels of IL-1β from supernatants of HT29 cells were quantified by ELISA using the human IL-1β ELISA KIT (MyBioSource, San Diego, CA, USA) following manufacturer’s instructions. Briefly, in order to remove the cell debris, cell supernatants were centrifuged at 4 °C for 20 min at 1000 rcf and diluted 1:10 with PBS. Samples were incubated during 90 min at 37 °C in the precoated 96-well strip plate. Then, Detection Solution A was added during 45 min at 37 °C and after that, Detection Solution B was incubated for 45 min at 37 °C. TMB Substrate Solution was then added at 37 °C during 15 min. Finally, Stop Solution was added and absorbance was measured at 450 nm with the microplate reader SpectaMax Plus 384 (Molecular Devices, San Jose, CA, USA).

### 2.10. Hematoxylin-Eosin Staining

Mice colonic tissues, paraffin-embedded in 5 µm sections, were stained with Hematoxylin-Eosin to analyze histology after the induction of chronic DSS-colitis. After deparaffinization and rehydration, slides were incubated with Hematoxylin 1:25 (Sigma-Aldrich, Madrid, Spain) during 3 min at room temperature. Then, ethanol-HCl 0.5% was added over 30 s and ammonium hydroxide 1% was also added over 30 s. Finally, aqueous eosin Y solution (Sigma-Aldrich, Madrid, Spain) diluted with glacial acetic acid 0.5% was added over 3 min at room temperature and dehydrated. Sections were visualized with a light microscope (Leica DMi8, L’Hospitalet de Llobregat, Spain) using LEICA LAS X software. To analyze the histology of the tissue, the parameters of Obermeier et al. [37] were used. Briefly, as detailed in Table 5, it consisted of a scale from 0 to 8 which represented the presence of erosion, the depth and surface extension lesions in the epithelium, as well as the degree of inflammatory infiltrate. The total histological score represented the sum of the epithelium and infiltration score (total score = E + I) [37].

### 2.11. Sirius Red Staining

Mice colonic tissues, paraffin-embedded in 5 µm sections, were stained with Sirius Red in order to analyze the collagen layer after induction of chronic DSS-colitis. Slides were deparaffinized, rehydrated and incubated with Fast Green (Sigma-Aldrich, Madrid, Spain) over 15 min at room temperature and with Sirius Red 0.1% (Sigma-Aldrich, Madrid, Spain)/Fast Green 0.04% over 30 min at room temperature. Finally, slides were dehydrated and observed with a light microscope (Leica DMDMi8, L’Hospitalet de Llobregat, Spain) using LEICA LAS X software. Collagen deposition, represented by red coloration, was quantified as mean red intensity per tissue area, using ImageJ (National Institutes of Health, Bethesda, MD, USA). In the graph, it is represented as % of red area. In order to measure the thickness of the collagen layer, ImageJ (National Institutes of Health, Bethesda, MD, USA) was also used. The measurement was performed in a blinded manner by an observer unaware of the corresponding group for each mouse.

### 2.12. Statistical Analysis

Data were expressed as mean ± SEM and were compared by a t-test for comparisons between two groups and by analysis of variance (one-way ANOVA) with Tukey post hoc correction or Kruskal–Wallis with Dunn’s post hoc correction where appropriate for multiple comparisons. Statistical significance was considered with a *p*-value < 0.05. Correlations from data obtained in human samples were analyzed using Spearman’s correlation coefficient.

## 3. Results

### 3.1. SUCNR1 Mediates Inflammasome Priming in intestinal Epithelial Cells

The treatment with succinate—as well as the treatment with the inflammatory cocktail and the combination of both—increased gene and protein expression of SUCNR1 in epithelial cells, although the statistical significance was obtained in combination for gene expression and in the cocktail for protein expression (Figure 1a). As expected, the treatment with the inflammatory cocktail induced a significant increase in the expression of *NLRP3, ASC* and *CASP1*, while the treatment with succinate alone failed to significantly modify their expression (Figure 1c). The coadministration of succinate and the cocktail significantly potentiated the expression of *NLRP3*, *IL1B* and *IL18* induced by the cocktail (Figure 1c). The transient silencing of *SUCNR1* (Figure 1b) did not modify the expression of any inflammasome component analyzed in basal conditions, but impaired the inflammasome priming, since there was a significant reduction in the expression *of NLRP3*, *IL1B, IL18* and *ASC* in siSUCNR1 cells treated with the cocktail or the combination compared with siRNA control cells (Figure 1c).

The analysis of the protein levels of pro-Caspase-1 revealed that the treatment with the cocktail induced a significant increase, whereas levels of this protein were significantly attenuated in siSUCNR1 cells treated with the cocktail and the combination (Figure 1d). The quantification of secreted IL-1β protein levels by ELISA showed a significant increase in HT29 cells treated with the inflammatory cocktail, which was potentiated with the combination of both cocktail and succinate. A significant reduction in secreted levels of IL-1β was obtained in both experimental conditions via the silencing of SUCNR1 (Figure 1e). 

Finally, as shown in Figure 1f, succinate levels were quantified and 130.4 ± 11.1 µM of this metabolite were detected in the supernatant of vehicle HT29 cells. The treatment with the cocktail failed to significantly modify levels of succinate which were also similar in siSUCNR1-transfected cells.

### 3.2. SUCNR1 Mediates Basal and LPS-Stimulated Inflammatory Pathways in Intestinal Epithelial Cells

The effect of LPS on SUCNR1 expression was analyzed in HT29 cells and results showed a significant increase in both gene and protein expression of this receptor after treatment with LPS during 24 h (Figure 2a). As shown in Figure 2b, LPS increased the ratio pNFкB/NFкB, while it failed to significantly modify the ratio pERK-1/2/ERK-1/2 compared with vehicle-treated cells. The silencing of SUCNR1 provoked, in basal conditions, a reduction in the phosphorylation of both NFкB and ERK-1/2, only significant for ERK-1/2 in siSUCNR1 vehicle-treated cells. These levels were similar to that detected in siSUCNR1 LPS-treated cells (Figure 2b). Next, we wanted to elucidate whether this effect on inflammatory pathways triggered alterations in the expression of pro-inflammatory cytokines. As expected, LPS treatment induced a significant increase in the mRNA expression of *IL1B*, *iNOS* and *IL6* while no changes in *TNF-a* were detected; these changes were significantly reduced in LPS-treated siSUCNR1 transfected cells compared with LPS-treated siCtrl cells (Figure 2b).

In order to analyze the effect of pERK-1/2 on NFкB activation, we treated HT29 cells over 2 h with LPS and the MEK inhibitor U0126. LPS treatment for 2 h did not modify the levels of pERK-1/2, while it reduced the protein levels of IкB in parallel, with a significant increase in pNFкB (Figure 2c). Of interest, the treatment with the MEK inhibitor U0126, impaired the LPS-reduction of IкB and significantly reduced the phosphorylated levels of NFкB (Figure 2c). 

Quantification of succinate levels in the supernatant revealed nonsignificant differences between any of the conditions analyzed (Figure 2d).

### 3.3. Lack of SUCNR1 Ameliorates DSS-Chronic Colitis

In order to analyze the relevance of SUCNR1 receptors in chronic colitis, we treated WT and SUCNR1^−/−^ mice with four cycles of increasing percentage of DSS in drinking water over 7 days intercalated with 10 days of water. At day 60, the survival proportion was significantly higher in SUCNR1^−/−^ mice with no deaths compared with WT mice which exhibited a 66.67% rate of survival (Figure 3a). In line with this, treatment with DSS induced a loss of body weight after four cycles which was significantly more pronounced in WT mice than in SUCNR1^−/−^ mice (Figure 3b). Although chronic administration of DSS induced a significant reduction in the colon length in both WT and SUCNR1^−/−^ mice (Figure 3c), the histology was more preserved and less altered in SUCNR1^−/−^ DSS-treated mice than in WT DSS-treated mice. Indeed, the histological analysis, performed following Obermeier et al. scale [37], demonstrated less infiltration and less affected epithelia in SUCNR1^−/−^ DSS-treated mice compared to WT-DSS treated mice (Figure 3d).

### 3.4. SUCNR1 Impairs the Expression of Inflammasome Components in DSS-Chronic Colitis

The role of SUCNR1 in the expression of the inflammasome components in vivo was examined in WT and SUCNR1^−/−^ mice after the induction of chronic colitis. The four cycles of DSS induced a significant increase in the mRNA expression of *Nlrp3*, *Casp1* and *Il1b* and protein levels of pro-Caspase-1 in colonic tissue, while no changes were observed in the *Asc* mRNA expression (Figure 4a,b). Interestingly, SUCNR1^−/−^ DSS-treated mice exhibited a significant reduction in the mRNA expression of *Nlrp3*, *Casp-1* and *Il1b* as well as in the protein expression of pro-Caspase-1 in the colon, compared with WT DSS-treated mice (Figure 4a,b).

### 3.5. SUCNR1 Deficiency Reduces Intestinal inflammation and Fibrosis in DSS-Chronic Colitis

The analysis of the expression of proinflammatory and anti-inflammatory cytokines in the colon revealed that chronic DSS treatment induced a significant increase in the expression of *Cox2*, *Tnf-a*, *iNos*, and *Il6* in WT mice, which was significantly reduced in SUCNR1^−/−^ DSS-treated mice. In knock-out mice treated with DSS, we detected a significant increase in the expression of the anti-inflammatory cytokine *Il10* compared with WT-DSS-treated mice (Figure 5a). 

Next, we studied the macrophage infiltration and macrophage phenotype in these mice. Chronic DSS treatment significantly increased the expression of *F4/80* only in WT mice, whereas nonsignificant differences were observed in SUCNR1^−/−^ DSS-treated mice (Figure 5b). In WT-DSS-treated mice, the expression of *Cd86*, *Ccr7*, *Arginase*, *Cd206* and *Cd16* were significantly increased. In contrast, in SUCNR1^−/−^ DSS-treated mice the increase in mRNA expression of *Cd86* was attenuated, while the increase in the mRNA expression of both Arginase and *Cd206* was potentiated in comparison with WT DSS-treated mice (Figure 5b).

Aside from inflammation, chronic DSS treatment induced a significant increase in the expression of profibrotic markers such as *Col1*, *Col3*, *Col4*, *Tgf-b*, *Timp1* and *Mmp2* in WT mice (Figure 5c). On the other hand, SUCNR1^−/−^ DSS-treated mice showed a significant reduction in the expression of *Col1*, *Col3*, *Col4*, *Vimentin*, *Tgf-b* and *Mmp2* compared with WT DSS-treated mice (Figure 5c). Finally, the percentage of red area and the analysis of the thickness of the collagen layer by Sirius Red staining revealed that WT DSS-treated mice presented a thicker collagen layer and increased red percentage area than those detected in SUCNR1^−/−^ DSS-treated mice (Figure 5d). 

### 3.6. SUCNR1 and Inflammasome Components Are Increased and Positively Correlate between Them in Surgical Resections from UC Patients

Finally, we analyzed the expression of SUCNR1 and inflammasome components in surgical resections obtained from UC patients. As shown in Figure 6a,b, gene and protein expression of SUCNR1 was significantly increased in UC patients compared with non-IBD patients. In parallel, the mRNA expression of NLRP3, CASP-1 and IL1B was significantly higher in samples from UC patients than in control tissue, whereas no differences were detected in the expression of ASC nor IL18 (Figure 6a). The ratio of Caspase-1/pro-Caspase-1 was significantly increased in resections from UC patients, as well as NLRP3 protein levels (Figure 6b). Next, we wanted to elucidate whether the succinate receptor was associated with the inflammasome activation in human intestinal mucosa of UC patients. Results revealed a positive and significant correlation between the expressions of *SUCNR1* and *NLRP3*, *CASP1*, *IL1B*, *IL18* and *ASC* (Figure 6c.)

## 4. Discussion

The present study demonstrated, for the first time, that SUCNR1 mediated the inflammasome priming and the NFкB-pathway activation in both intestinal epithelial cells and a chronic murine model of colitis, suggesting a role for this receptor in Ulcerative Colitis. 

Our data showed that SUCNR1 mediated the first step of the inflammasome activation in IECs, since the transient silencing of this receptor reduced the gene expression of all inflammasome components and the protein levels of both pro-Caspase-1 and IL-1β. Epithelial cells are the first line of defense against pathogens and several intestinal insults, and the inflammasome activation may play a pivotal defensive role [38]. Most studies have focused on the role of the inflammasome activation in epithelial cells. For instance, it was reported that the inflammasome activation prevented *Staphylococcus aureus* infection in the eye [39] or influenza A viral infection in human airway epithelial cells [40]. In addition, the activation of inflammasome regulated intestinal healing in DSS-treated mice, as demonstrated by the use of Caspase-1 knock-out mice [41]. Nevertheless, there is a lack of studies focused specifically on the molecular pathways involved in the priming of the inflammasome. The present study revealed, for the first time, that SUCNR1 was able to prime the NLRP3 inflammasome in intestinal epithelial cells, which suggested the involvement of this receptor in the inflammasome activation and, subsequently, in the maintenance of intestinal mucosal integrity. 

Of interest, our results also showed that SUCNR1 mediated a constitutive regulation of the ERK-1/2 pathway in intestinal epithelial cells, since siSUCNR1-cells exhibited a significant reduction in levels of pERK-1/2, in line with previous studies in endothelial cells [42] and in cells from the retinas of diabetic rats [43]. The only known ligand of SUCNR1 is the TCA metabolite succinate; we propose that the constitutive activation of this G-protein-coupled receptor might be mediated by the high levels of succinate detected in the epithelial cellular supernatant. In fact, considering that the reported EC50 of succinate was 91 ± 14 µM [44], the levels that we detected in the supernatant of HT29 cells were more than enough to reach the maximum activation of SUCNR1 in basal conditions. Moreover, the present data extend these observations by showing that SUCNR1 regulated the basal phosphorylation of the well-known proinflammatory transcription factor NFкB in intestinal epithelial cells (Figure 7). These observations may help to explain the reduced expression of proinflammatory cytokines detected in resting macrophages by several studies [26,30]. Growing evidence has demonstrated that NFкB induces the transcription of several inflammasome components such as Nlrp1, Nlrp3, and IL-1B [45]. Taking this into account, we suggest that SUCNR1 regulates the inflammasome priming through NFкB since the transient silencing of SUCNR1 reduces the phosphorylation of this transcription factor and, subsequently, the transcription of the inflammasome components, as shown in Figure 7.

Previous studies have specifically located SUCNR1 in the apical membrane of renal and retinal epithelial cells [44,46]. Considering that epithelial cells are in close contact with the intestinal microbiota, a source of succinate [47] and bacterial toxins, it seems likely that the activation of proinflammatory pathways by SUCNR1 plays a homeostatic role in the maintenance of mucosal integrity. In this line, we found that the increased phosphorylation of NFкB and the consequent transcription of IL1B, iNOS and IL6, induced by 24 h LPS-treatment, were significantly prevented in cells in which SUCNR1 had been knocked down. Unexpectedly, LPS failed to modify levels of pERK-1/2, even when these cells were analyzed 2 h after LPS treatment, conditions in which we did detect a reduction of IкB levels and an increase of pNFкB. It seemed that LPS activated inflammatory pathways by ERK-1/2-independent mechanisms; however, the use of the ERK-1/2 inhibitor U0126, which abolished the ability of these cells to phosphorylate this kinase, increased IкB levels and reduced the phosphorylated levels of NFкB, in LPS-treated cells. Taken together, our results strongly suggested that activation of NFкB in both basal and LPS-stimulated conditions depended on the constitutive cellular levels of pERK-1/2 (Figure 7). 

Both LPS and the inflammatory cocktail upregulated the gene and protein expression of SUCNR1 in intestinal epithelial cells, in line with that reported in bone marrow-derived macrophages (BMDMs) [48]. Unexpectedly, they both failed to significantly increase the extracellular levels of succinate, in contrast to that reported in BMDMs [30]. It is important to note that differences in succinate levels between cell lines and primary cells have been reported [42]; we found that, in HT29 cells, levels of this metabolite were on the order of ten times higher than those detected in blood from carcinoma [49] or Crohn’s Disease patients [26]. Therefore, we suggest that, in intestinal epithelial HT29 cells, SUCNR1 (rather than succinate levels) might be the limiting factor for the activation of the proinflammatory signalling pathway. Reinforcing this observation, treatment with 1 mM of succinate per se failed to activate inflammasome, while it potentiated the activation of this process by a specific cocktail—probably due to the increased SUCNR1 protein expression associated to that condition. 

The functional role of SUCNR1 in vivo was analyzed in a chronic murine model of colitis induced by DSS. In line with previous studies showing a role for NLRP3 inflammasome in DSS colitis etiopathogenesis [14,15], we found an increased expression of both IL1B mRNA and inflammasome components, as well as protein levels of pro-Caspase-1, in colonic tissue of DSS-treated mice. The lack of SUCNR1 by the use of SUCNR1^−/−^ mice significantly prevented the expression of inflammasome components induced by chronic DSS. Indeed, SUCNR1^−/−^ mice exhibited ameliorated chronic colitis and a higher survival, compared with WT mice. Previous studies have reported a role for SUCNR1 in acute colitis [26], arthritis [30,50,51], isoproterenol-induced myocardial ischemia [52], obesity and diabetes [31], but as far as we know, the present study was the first to show a role for this receptor in the in vivo regulation of inflammasome priming. The absence of SUCNR1 was also associated with a reduction in the expression of proinflammatory cytokines and M1 markers and a higher expression of M2 markers such as *Cd206* and *Arginase*, in colonic tissue of DSS-treated mice. The attenuation of the inflammatory process ran in parallel with a significant reduction in the expression of the most common profibrotic markers such as *Col1*, *Col3*, *Col4*, *Tgf-b*, *Vimentin* and a reduced thickness of collagen layer. Data in the present study reinforced previous observations reported in intestinal fibrosis using the heterotopic transplant model [53], and extended them by showing that the absence of SUCNR1 impaired the inflammasome priming and intestinal fibrosis induced by chronic DSS treatment. SUCNR1 was detected in several cell types involved in fibrosis development, such as fibroblasts, epithelial cells and immune cells [27], and present and previous studies have demonstrated a proinflammatory role of SUCNR1 in all of them. It seems likely that, in chronic conditions of intestinal inflammation, the proinflammatory role of SUCNR1 in immune cells [30] prevailed over the homeostatic mechanism mediated by this receptor in epithelial cells, some of which may have been lost in the affected tissue. Reinforcing this observation, our study showed a higher expression of SUCNR1 and inflammasome activation in colonic resections from UC patients, in line with that previously reported in the ileum of Crohn´s Disease patients [26]. Of interest, a positive and significant correlation was detected between SUCNR1 and several inflammasome components, such as *NLRP3*, *CASP-1*, *IL1B*, *IL18* and *ASC* expression. Taken together, our results suggested that blockading SUCNR1 could benefit patients with an UC outbreak, but the dosage schedule must consider the protective role played by this receptor in the maintenance of epithelial barrier function. 

In summary, our study demonstrated a role for SUCNR1 in epithelial activation of inflammatory pathways, suggesting a homeostatic role for this receptor in the maintenance of intestinal integrity. However, this receptor is also involved in the first step of the inflammasome associated with chronic murine and human colitis. Additional studies are needed before proposing this receptor as a new pharmacological target in the treatment of chronic inflammatory conditions.

## Figures and Tables

**Figure 1 biomedicines-10-00532-f001:**
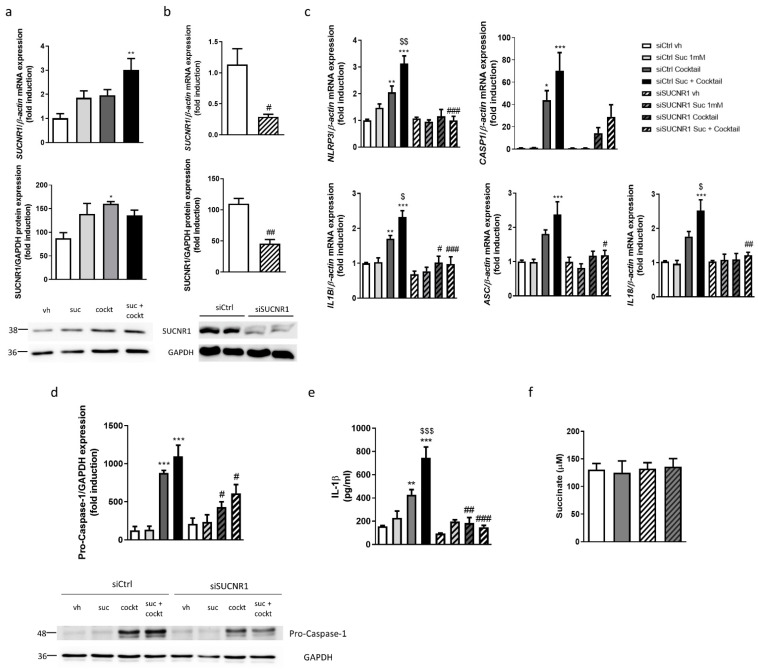
SUCNR1 mediates inflammasome priming in intestinal epithelial cells. (**a**) HT29 cells treated with succinate 1 mM and/or an inflammatory cocktail. Graphs show mRNA (*n* = 7) and protein (*n* = 3) expression of SUCNR1. Image of a representative Western Blot of one independent experiment. Bars in graphs represent mean ± SEM; * *p* < 0.05 and ** *p* < 0.01 vs. vehicle cells. (**b**–**e**) HT29 cells transiently transfected with a specific siRNA against SUCNR1 or ctrl; (**b**) Graphs show mRNA and protein expression of SUCNR1 (*n* = 5). (**c**) Graphs show mRNA expression of *NLRP3, CASP1*, *IL1B*, *ASC* and *IL18* (*n* = 5). (**d**) Graph shows protein expression of pro-Caspase-1 (*n* = 4). Image of a representative Western Blot of one independent experiment. (**e**) Graph shows secreted protein levels of IL-1β detected in the supernatant of HT29 cells (*n* = 3). (**f**) Graph shows succinate levels in supernatant of HT29 cells (*n* = 6). In all cases, bars in graphs represent mean ± SEM. * *p* < 0.05, ** *p* < 0.01 and *** *p* < 0.001 vs. siCtrl vehicle cells. # *p* < 0.05, ## *p* < 0.01 and ### *p* < 0.001 vs. the respective siCtrl cells. $ *p* < 0.05, $$ *p* < 0.01, $$$ *p* < 0.001 vs. siCtrl cocktail treated cells.

**Figure 2 biomedicines-10-00532-f002:**
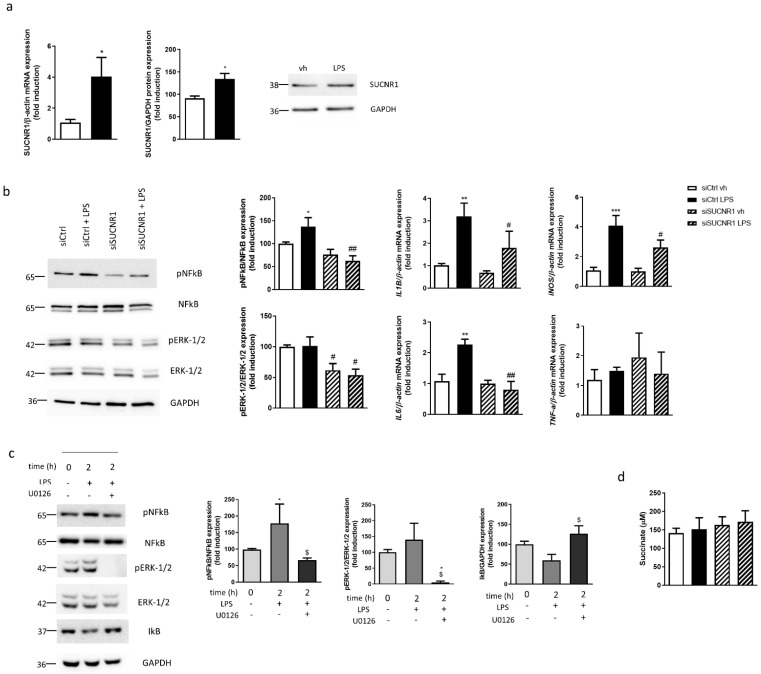
SUCNR1 mediates basal and LPS-stimulated inflammatory pathways in intestinal epithelial cells. (**a**) HT29 cells treated with LPS 0.1 µg/mL during 24 h and graphs show mRNA and protein expression of SUCNR1 (*n* = 5). Image of a representative Western Blot of one independent experiment. (**b**) HT29 cells transiently transfected with a specific siRNA against SUCNR1 or ctrl and treated with LPS 0.1 µg/mL during 24 h. Graphs show protein expression of pERK-1/2, ERK-1/2, pNFкB and NFкB (*n* = 6) and mRNA expression of *IL1B*, *iNOS*, *IL6* and *TNF-a* (*n* = 5). Image of a representative Western Blot of one independent experiment. (**c**) HT29 cells treated with vehicle or LPS (with or without the MEK inhibitor U0126 10 µM) over 2 h and graphs show protein expression of pERK-1/2, ERK-1/2, pNFкB, NFкB and IкB (*n* = 5). Image of a representative Western Blot of one independent experiment. (**d**) Graph shows succinate levels in supernatant of HT29 cells (*n* = 4). In all cases, bars in graphs represent mean ± SEM. * *p* < 0.05, ** *p* < 0.01 and *** *p* < 0.001 vs. vehicle cells. # *p* < 0.05 and ## *p* < 0.01 vs. the respective siCtrl cells. $ *p* < 0.05 vs. U0126 nontreated cells.

**Figure 3 biomedicines-10-00532-f003:**
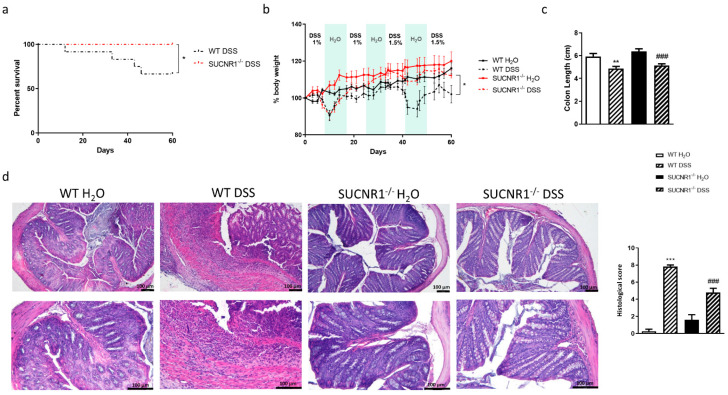
Lack of SUCNR1 ameliorates chronic DSS-chronic colitis. Chronic intestinal colitis was induced in vivo in WT and SUCNR1^−/−^ mice with four cycles of increasing percentage of DSS (1%, 1%, 1.5% and 1.5%) in drinking water over 7 days, intercalated with 10 days of water. At the end of the last cycle, on day 60, mice were euthanized and colon tissue samples were collected. (**a**) Graph shows the survival percentage in mice after the four cycles of DSS (*n* = 10). (**b**) Graph shows the evolution of body weight (*n* = 10). (**c**) Graph shows the colon length of mice (*n* = 10). (**d**) Hematoxylin-Eosin staining performed on intestinal resections of WT and SUCNR1^−/−^ mice and representative pictures of each group are shown (*n* = 10). Histological score to assess the integrity of the epithelium and the degree of infiltration also performed following Obermeier et al. parameters Bars in graphs represent mean ± SEM. * *p* < 0.05, ** *p* < 0.01 and *** *p* < 0.001 vs. WT H_2_O mice. ### *p* < 0.0001 vs. WT DSS-treated mice.

**Figure 4 biomedicines-10-00532-f004:**
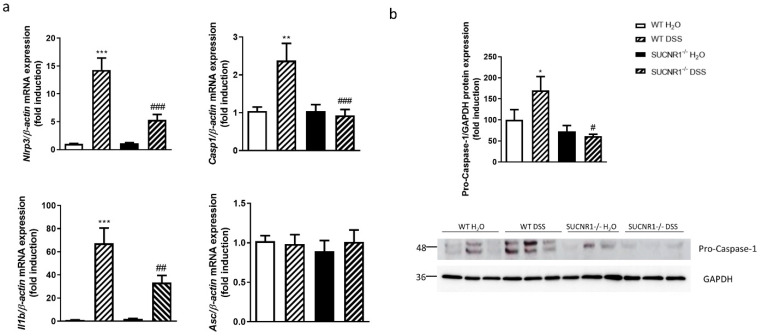
SUCNR1 impairs the expression of inflammasome components in DSS-chronic colitis. Chronic intestinal colitis was induced in vivo in WT and SUCNR1^−/−^ mice with four cycles of increasing percentage of DSS in drinking water over 7 days, intercalated with 10 days of water. At the end of the last cycle, on day 60, mice were euthanized and colon tissue samples were collected. (**a**) Graphs show mRNA expression of *Nlrp3*, *Casp*-1, *Il1b* and *Asc* (*n* = 8). (**b**) Graph shows protein expression of pro-Caspase-1 (*n* = 3). Bars in graphs represent mean ± SEM. * *p* < 0.05, ** *p* < 0.01 and *** *p* < 0.001 vs. WT H_2_O mice. # *p* < 0.05, ## *p* < 0.01 and ### *p* < 0.0001 vs. WT DSS-treated mice.

**Figure 5 biomedicines-10-00532-f005:**
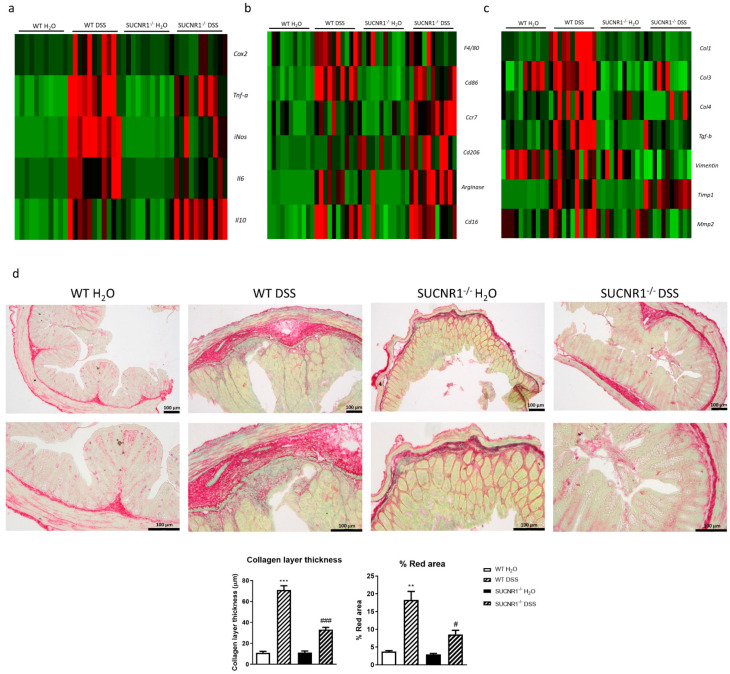
SUCNR1 deficiency reduces intestinal inflammation and fibrosis in DSS-chronic colitis. Intestinal colitis was induced in vivo in WT and SUCNR1^−/−^ mice with four cycles of increasing percentage of DSS in drinking water over 7 days, intercalated with 10 days of water. At the end of the last cycle, on day 60, mice were euthanized and colon tissue samples were collected. (**a**) Heat map showing the mRNA expression of proinflammatory and anti-inflammatory cytokines including *Cox2, Tnf-a, iNos, IL6* and *Il10*. (**b**) Heat map showing the mRNA expression of macrophage infiltration and phenotype markers such as *F4/80, Cd86, Ccr7, Cd206, Arginase* and *Cd16*. (**c**) Heat map showing the mRNA expression of profibrotic markers including *Col1, Col3, Col4, Tgf-b, Vimentin, Timp1* and *Mmp2*. (**d**) Sirius-Red staining performed on intestinal resections of WT and SUCNR1^−/−^ mice and representative images of each group (*n* = 8). In addition, quantification of the collagen layer thickness as well as % of red area is also represented. Bars in graphs represent mean ± SEM. ** *p* < 0.01 and *** *p* < 0.001 vs. WT H_2_O mice. # *p* < 0.05 and ### *p* < 0.001 vs. WT DSS-treated mice.

**Figure 6 biomedicines-10-00532-f006:**
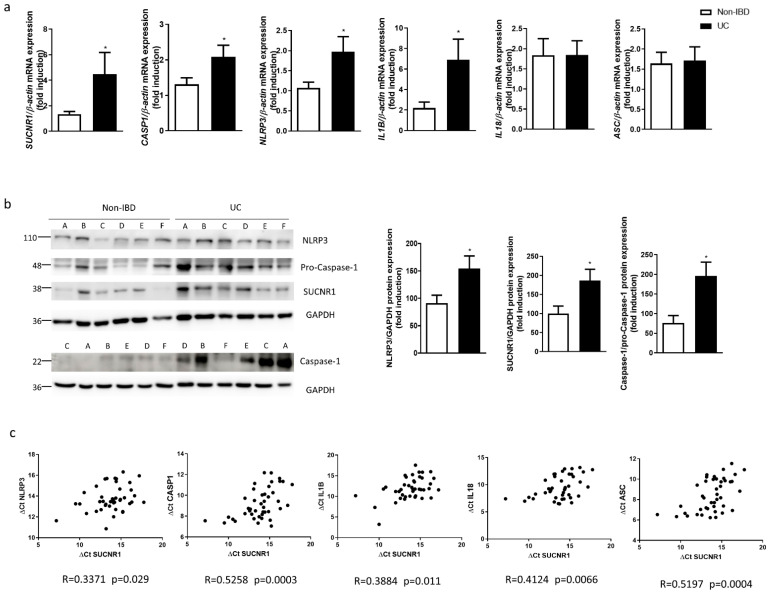
SUCNR1 and inflammasome components are increased and positively correlated in surgical resections from UC patients. (**a**) Graphs show mRNA expression of *SUCNR1*, *NLPR3*, *CASP-1*, *IL1B*, *IL18* and *ASC* in intestinal resections from UC patients (*n* = 25) and non-IBD patients (*n* = 30). (**b**) Graphs show protein expression of SUCNR1, NLRP3 and Caspase-1/pro-Caspase-1 (*n* = 15). Bars in graphs represent mean ± SEM. * *p* < 0.05 vs. non-IBD patients. (**c**) Graphs show the correlations between data relative to mRNA expression of SUCNR1 vs. mRNA expression of the inflammasome components *NLRP3*, *CASP1*, *IL1B* and *IL18* (expressed as ΔCt). In each correlation, the value of the Spearman’s correlation coefficient (R) and *p* value are shown.

**Figure 7 biomedicines-10-00532-f007:**
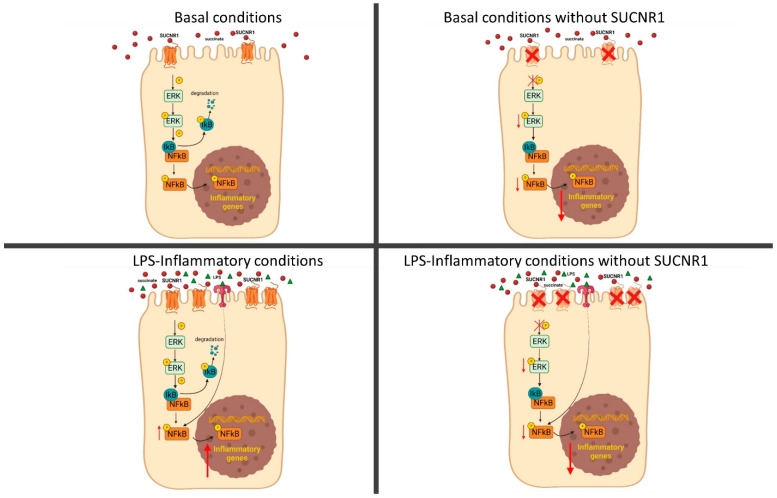
The activation of SUCNR1 by constitutive levels of succinate regulates ERK and NFкB phosphorylation in non-stimulated epithelial cells. In LPS-treated cells, there is an upregulation of SUCNR1 which increases the phosphorylation of ERK and NFкB, with the subsequent expression of proinflammatory genes.

**Table 1 biomedicines-10-00532-t001:** Information about UC and Non-IBD patients.

	Ulcerative Colitis Patients	Non-IBD Patients
**Number of patients**	25	30
**Age**		
**Median**	47	54
**Interval**	(17–69)	(18–85)
**Gender**		
**Male**	13 (52%)	18 (60%)
**Female**	12 (48%)	12 (40%)
**Localization**		
**Pancolitis**	10 (40%)	
**Distal Colitis**	5 (20%)	
**Proctosigmoiditis**	10 (40%)	
**Treatment**		
**Azathioprine**	4	
**Corticoids**	7	
**Anti-Inflammatory Drugs**	6	
**Mesalazine**	6	
**Budesonide**	1	

**Table 2 biomedicines-10-00532-t002:** Primary antibodies used for Western Blot analysis.

Antibody	Supplier	Dilution
NLRP3	13158, Cell Signaling	1:1000
Caspase-1	2225, Cell Signaling	1:1000
Cleaved Caspase-1 (Asp297)	4199, Cell Signaling	1:1000
Phospho-NFкB	3033S, Cell Signaling	1:1000
NFкB	8242S, Cell Signaling	1:1000
Phospho-ERK	9104, Cell Signaling	1:1000
ERK	4695, Cell Signaling	1:1000
IкB	SC-371, Santa Cruz Biotechnology	1:1000
SUCNR1	IMG-6352a, IMGENEX	1:1000
GAPDH	G9545, Sigma-Aldrich	1:5000

**Table 3 biomedicines-10-00532-t003:** Sequences of human primers used in real-time PCR.

Gene	Sense (5′-3′)	Antisense (3′-5′)	Fragment’s Size (bp)
*IL1B*	GCTCGCCAGTGAAATGATGG	TCGTGCACATAAGCCTCGTT	330
*iNOS*	ATAATGGACCCCAGGCAAG	TCAGCAAGCAGCAGAATGAG	195
*IL6*	AGTGAGGAAGCCAGAGC	ATTGTGGTTGGGTCAGGGG	143
*TNF* *-a*	GCTGCACTTTGGAGTGATCG	GGGTTTGCTACAACATGGGC	138
*NLRP3*	AGAACTGTCATCGGGTGGAG	AACTGGAAGTGAGGTGGCTG	174
*ASC*	CAAACGTTGAGTGGCTGCTG	GAGCTTCCGCATCTTGCTTG	107
*CASP1*	AGAGAAAAGCCATGGCCGAC	CCTTCACCCATGGAACGGAT	70
*IL18*	GCTGAAGATGATGAAAACCTGGA	GAGGCCGATTTCCTTGGTCA	115
*SUCNR1*	GGAGACC CCAACTACAACCTC	AGCAACCTGCCTATTCCTCTG	132
*β-actin*	GGACTTCGAGCAAGAGATGG	AGCACTGTGTTGGCGTACAG	57

**Table 4 biomedicines-10-00532-t004:** Sequences of mouse primers used in real-time PCR.

Gene	Sense (5′-3′)	Antisense (3′-5′)	Fragment’s Size (bp)
*Cox-2*	CCCGGACTGGATTCTATGGTG	TTCGCAGGAAGGGGATGTTG	153
*Tnf-* *a*	GATCGGTCCCCAAAGGGATG	GGTGGTTTGTGAGTGTGAGGG	86
*iNos*	CGCTTGGGTCTTGTTCACTC	GGTCATCTTGTATTGTTGGGCTG	222
*Il6*	ATGAGGAGACTTGCCTGGTG	CTGGCATTTGTGGTTGGGTC	202
*Il10*	GGACAACATACTGCTAACCGAC	CCTGGGGCATCACTTCTACC	110
*F4/80*	TGTCTGAAGATTCTCAAAACATGGA	TGGAACACCACAAGAAAGTGC	211
*Cd86*	GCACGGACTTGAACAACCAG	CCTTTGTAAATGGGCACGGC	194
*Ccr7*	CTCTCCACCGCCTTTCCTG	ACCTTTCCCCTACCTTTTTATTCCC	125
*Arginase*	GTGGGGAAAGCCAATGAAGAG	TCAGGAGAAAGGACACAGGTTG	232
*Cd16*	GAAGGGGAAACCATCACGCT	GCAAACAGGAGGCACATCAC	293
*Col1*	CAGGCTGGTGTGATGGGATT	AAACCTCTCTCGCCTCTTGC	317
*Col3*	CTACACCTGCTCCTGTGCTTC	GATAGCCACCCATTCCTCCCA	237
*Col4*	ATTAGCAGGTGTGCGGTTTG	ATTAGCAGGTGTGCGGTTTG	289
*TGF* *β*	GCGGACTACTATGCTAAAGAGG	TCAAAAGACAGCCACTCAGG	295
*Vimentin*	GCTCCTACGATTCACAGCCA	CGTGTGGACGTGGTCACATA	190
*Timp1*	GGCATCTGGCATCCTCTTGTTG	GTGGTCTCGTTGATTTCTGGGG	147
*Mmp2*	GCCAACTACAACTTCTTCCCC	CAAAAGCATCATCCACGGTTTC	112
*Nlrp3*	GTACCCAAGGCTGCTATCTGG	TGCAACGGACACTCGTCATC	143
*Asc*	TGACTGTGCTTAGAGACATGGG	AACTGCCTGGTACTGTCCTTC	233
*Casp1*	CTCGTACACGTCTTGCCCTC	GGTCCCACATATTCCCTCCTG	260
*Il18*	GCTTGCTTTCACTTCTCCCC	TGCCTGGATGCTTGTAAACTTG	262
*Sucnr1*	GACAGAAGCCGACAGCAGAATG	GCAGAAGAGGTAGCCAAACACC	160
*β-actin*	GCCAACCGTGAAAAGATGACC	GAGGCATACAGGGACAGCAC	95

**Table 5 biomedicines-10-00532-t005:** Histological score parameters.

	Epithelium (E)		Infiltration (I)
0	Normal morphology	0	No infiltrate
1	Loss of epithelial cells	1	Infiltrate around crypt basis
2	Loss of epithelial cells in large areas	2	Infiltrate reaching to L. muscularis mucosae
3	Loss of crypts	3	Extensive infiltration reaching the L. muscularis mucosae and thickening of the mucosa with abundant oedema
4	Loss of crypts in large areas	4	Infiltration of the L. submucosa

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
