# Peer review of "SUCNR1 Mediates the Priming Step of the Inflammasome in Intestinal Epithelial Cells: Relevance in Ulcerative Colitis"

_biomedicines, 2022, doi:10.3390/biomedicines10030532_

Round 1

Reviewer 1 Report

The authors investigated the influence of SUCNR1 in ulcerative colitis. They did analyses with cell cultures of HT29 cells, the experimental DSS colitis model and resections of colitis patients. They found that SUCNR1 plays a crucial role in both the cells and the colitis model. The UC patients showed increased expression of the receptor during inflammation phase and the inflammasome components. Ultimately, the authors suggest a more prominent role of the receptor and therefore propagating for a better understanding of inflammatory activity for UC.

Overall, the manuscript was undestandable and the experiments are well done. The experiments are logically structured and comprehensible. There are just a few things I noticed that should be improved.

In figure 1B, a marker or arrows should be added to make it easier to understand how the cycles of the DSS application were given. Otherwise the weight curve is difficult to interpret. In Figure 1D, the scale bar or size scale is missing in 3 partial figures. A corresponding scale for the histological inflammations is also missing. There are several sources in the literature avaible ( like Erben et al., Int. J. Clin Exp. Pathol 2014, 7(8), 4557) on how to create the parameters. It would be nice if a corner of the figure could show an enlargement or magnification of the section so that the cell structure can be better interpreted.

Concerning the qPCR analyses, GAPDH was always used as the housekeeping gene. Why this gene in particular was choosen and not, for example, beta-actin or 18sR-RNA ? An explanation would be helpful for the readers.

Figure 5D also shows histological sections of fibrinous colon sections and also lacks a score for the thickness of the fibrin layer. There are also parameters in the literature avaible especially for Sirius Red staining samples to assess the severity of the thickening and to translate to a score.

Otherwise, everything is fine and I would suggest a minor revision.

Author Response

The authors investigated the influence of SUCNR1 in ulcerative colitis. They did analyses with cell cultures of HT29 cells, the experimental DSS colitis model and resections of colitis patients. They found that SUCNR1 plays a crucial role in both the cells and the colitis model. The UC patients showed increased expression of the receptor during inflammation phase and the inflammasome components. Ultimately, the authors suggest a more prominent role of the receptor and therefore propagating for a better understanding of inflammatory activity for UC.

Overall, the manuscript was undestandable and the experiments are well done. The experiments are logically structured and comprehensible. There are just a few things I noticed that should be improved.

In figure 1B, a marker or arrows should be added to make it easier to understand how the cycles of the DSS application were given. Otherwise the weight curve is difficult to interpret.

We appreciate so much the comment and according referee`s suggestion we have now modified the Figure 3B in order to make easier the interpretation of the chronic DSS protocol administration.

In Figure 1D, the scale bar or size scale is missing in 3 partial figures. A corresponding scale for the histological inflammations is also missing. There are several sources in the literature avaible ( like Erben et al., Int. J. Clin Exp. Pathol 2014, 7(8), 4557) on how to create the parameters. It would be nice if a corner of the figure could show an enlargement or magnification of the section so that the cell structure can be better interpreted.

According referee`s suggestion we have now included the scale bar in all the pictures of Figure 3D. In addition, in this version of the manuscript we have also quantified the histological inflammation following Obermeier F., 1999; doi: 10.1046/j.1365-2249.1999.00878.x. score which represents the sum of the epithelium and infiltration score, which has also been described in Materials and methods. Hence, a graph showing the results has been added to Figure 3D. Moreover, following referee´s comments, a magnification of the section in each figure has also been added to the complete figure so as to have a deeper view of the cell structure.

Concerning the qPCR analyses, GAPDH was always used as the housekeeping gene. Why this gene in particular was choosen and not, for example, beta-actin or 18sR-RNA ? An explanation would be helpful for the readers.

We appreciate reviewer´s suggestion, but, as indicated in Materials and methods 2.7 RNA isolation and Real-Time Quantitative PCR (RT-qPCR), beta-actin was used as housekeeping gene in qPCR analysis while GAPDH was used as housekeeping in protein analysis.

Figure 5D also shows histological sections of fibrinous colon sections and also lacks a score for the thickness of the fibrin layer. There are also parameters in the literature avaible especially for Sirius Red staining samples to assess the severity of the thickening and to translate to a score.

We fully agree with the reviewer’s comment, and we have now added a quantification of the thickness of the collagen layer performed using ImageJ to determine the collagen thickness layer as well as the % of red area. Therefore, in this version of the manuscript we have added the graphs in Figure 5D and the methodology followed in Materials and methods. Indeed, we have also added the scale bar in all figures, as well as another image with higher magnification for each group.

Otherwise, everything is fine and I would suggest a minor revision.

Reviewer 2 Report

Please see the reviewer's comment attached.

Reviewer 3 Report

Dear Authors,

  1. A minor proofreading of the text is needed.
  2. Materials and methods: 2.3. Introduction of experimental DSS colitis in mice – How many groups did you use? How many mice were in each group?
  3. Results; Figure 2c – You presents three bars and x-axis is signed time (h) 0, 2 and 2, is not supposed to be 0, 1 and 2?

Author Response

1. A minor proofreading of the text is needed.

We appreciate reviewer’s suggestion and we have done a deeper proofread of the text.

2. Materials and methods: 2.3. Introduction of experimental DSS colitis in mice – How many groups did you use? How many mice were in each group?

Following reviewer’s suggestion, we have now better described how many groups we used and how many mice were in each group.

3. Results; Figure 2c – You presents three bars and x-axis is signed time (h) 0, 2 and 2, is not supposed to be 0, 1 and 2?

We thank reviewer’s comment but the signed time is in fact 0, 2 and 2 because at 2 hours we have analysed the effects with and without the ERK inhibitor U0126.

Round 2

Reviewer 2 Report

The manuscript improved significantly. But this reviewer still feels lack of primary cell's data along with HT29. Human primary HIEC-6 cell is available at ATCC and this reviewer used it recently.

Author Response

The manuscript improved significantly. But this reviewer still feels lack of primary cell's data along with HT29. Human primary HIEC-6 cell is available at ATCC and this reviewer used it recently.

We highly appreciate the comment of the referee regarding the use of the human primary HIEC-6 cells. We strongly believe that his/her suggestion would significantly improve the quality of the manuscript given the fact that those cells are a primary non-cancerous cell line, unlike HT29 cells. Nevertheless, unfortunately, we have not these cells in our laboratory at the moment and we have never worked with them. So, if we need to order, expand them and replicate all the experiments performed in this manuscript, we would need some months to achieve that issue. In addition, to our knowledge, there is no research work available in literature describing HIEC-6 transfection with Lipofectamine, the technique used in our research work as described in Materials and methods (2.5). Hence, up to now, the viability of cells after performing this technique is not widely studied, which would force us to perform several trials in order to optimize the protocol of transfection.

On the other hand, as indicated in the datasheet of HIEC-6, this cell line specifically corresponds to the small intestine, described as enteric intestinal crypt cells. Thus, in this case, given the fact that all the patients included in the study suffer Ulcerative Colitis and our human controls were taken from the healthy mucosa of coloncarcinoma patients, we consider that the use of HT29 cells, instead of HIEC-6 cells, is adequate.

Taking everything into account, we would like to highlight that reviewer’s suggestion is really valuable for us and we will absolutely consider this option in our ongoing and future experiments. We honestly appreciate his/her recommendation but we do not think that the use of HIEC-6 cells is extremely necessary in this manuscript and it will not provide new conclusions from those already mentioned in the second version of the manuscript.